# Maternal dietary practices, dietary diversity, and nutrient composition of diets of lactating mothers in Jimma Zone, Southwest Ethiopia

**Sirawdink Fikreyesus Forsido**[1,2]*, **Frehiwot Tadesse**[2], **Tefera Belachew**[3], **Oliver Hensel**[1]

**1** Department of Agricultural and Biosystems Engineering, Faculty of Organic Agricultural Sciences, University of Kassel, Nordbahnhofstraße 1a, Witzenhausen, Germany, **2** Department of Post-Harvest Management, Jimma University, Jimma, Ethiopia, **3** Department of Population and Family Health, College of Health Sciences, Jimma University, Jimma, Ethiopia

* sirawdink.fikreyesus@ju.edu.et

## Abstract

### Background

Optimal nutrition during lactation is essential for the well-being of the mother and the infant. Studies have shown that access to nutrient-rich foods during lactation is critical as minimal stores of nutrients can have adverse effects. This study aimed to investigate the diversity, composition, and nutrient adequacy of diets of lactating mothers in Southwest Ethiopia.

### Methods

A community-based cross-sectional survey was carried out in three districts of Jimma Zone, Southwest Ethiopia, in February 2014. A stratified multistage sampling technique was used to select 558 lactating mothers. Data were collected using a pre-tested and structured interviewer-administered questionnaire. Minimum dietary diversity for women (MDD-W) was computed from a single 24-h recall. A cut off value of 5 was used to classify the dietary diversity into achieving MDD-W or not. The proximate, mineral and anti-nutritional compositions of 12 commonly consumed foods were analysed using standard methods. Nutrient adequacy ratio (NAR) and Mean adequacy ratio (MAR) of these foods were estimated.

### Results

The mean (±SD) dietary diversity score (DDS) of the study participants was 3.73±1.03. Meeting MDD-W was positively associated with agricultural production diversity (P = 0.001) and educational level of the women (P = 0.04). Conversely, district of the study (P = 0.003) and place of residence (P = 0.019) were negatively associated with meeting MDD-W. The proximate composition (g/100g) of the sampled foods ranged between 24.8–65.6 for moisture, 7.6–19.8 for protein, 2.1–23.1 for crude fat, 2.0–27 for crude fibre, 1.0–21.2 for total ash, and 0.9–45.8 for total carbohydrate content. The calorific value ranged between 124.5–299.6 Kcal/100g. The mineral contents (mg/100g) ranged between 9.5–52.5 for iron, 2.2–4.2 for zinc, 42.6–318.2 for calcium, and 150.7–379.9 for phosphorus. The content of anti-nutritional factors (mg/100g) ranged between 11.1–178.9 for phytate and 3.7–315.9 for

**Data Availability Statement:** We have uploaded our study's minimal underlying data set as a stable,

public repository and included the relevant DOI
(https://doi.org/10.5061/dryad.nk98sf7sr).

**Funding:** This research was funded by the
Bundesministerium für Bildung und Forschung,
Grant/Award Number: 031A247A.

**Competing interests:** We have no conflicts of
interest to declare.

tannin. All the commonly consumed maternal foods were not sufficient to meet the energy, fat and protein requirements, (NAR<1). However, all diets provided adequate iron and most of the cereal-based foods provided adequate carbohydrate and minerals. The overall nutrient adequacy was below the cut-off point for all food types.

## Conclusions

The diets of lactating mothers in Southwest Ethiopia lack diversity and nutrient adequacy. A community-based nutrition education program on the importance of diet diversity and nutrient intake during lactation based on a multi-sectoral approach is needed.

## Introduction

Maternal undernutrition is a significant cause of morbidity and mortality in low- and middle-income countries throughout the world and is estimated to affect 22% of the mothers of Ethiopia. Ethiopia is still among countries with a high burden of maternal malnutrition [1]. Mothers from low-income settings are amongst the most vulnerable to undernutrition because their body's nutrient reserve is severely affected due to frequent pregnancies and lactation [2]. Lactating women who do not get enough energy and nutrients in their diets risk maternal depletion [3–5].

According to the United Nations Children's Fund (UNICEF) conceptual framework of undernutrition, inadequate dietary intake is an immediate cause of maternal and child undernutrition [6]. Based on nutrition recommendations, energy needs are 25% higher during lactation, and protein needs are 54% higher during lactation compared to a non-lactating woman [7, 8]. However, lactating women from low-income settings are considered as a nutritionally vulnerable group because their dietary practices may not be optimal. Ethiopia experiences one of the highest incidences of child and maternal nutritional deficiencies which contribute to increased morbidity and mortality [9]. A malnourished mother will give birth to a low birth weight baby; the low birth weight baby will grow as a malnourished child, then to a malnourished teenager, then to a malnourished pregnant woman, and so the cycle continues [10].

One of the available tools for measuring dietary adequacy among lactating mothers is dietary diversity [11]. However, literature evidenced that dietary diversity among lactating mothers is very low in low-income settings [12]. Several demographic, socio-economic, and other factors, which could vary from setting to setting, could contribute to reduced dietary diversity [13].

Another gap that is noticed regarding maternal diets in low-income settings like Ethiopia is the composition of the foods customarily consumed is not exhaustively documented in the food composition tables. This information is fundamental for programmes which operate in the agriculture-nutrition-health interface. Inadequate food composition data may lead to a failure in understanding the relationship between nutrient intake and health or result in inappropriate, inefficient interventions to improve micronutrient status [14].

Information on dietary practices, dietary diversity and associated factors, and composition of foods of lactating mothers are urgently needed for prioritizing, designing and initiating intervention programs aiming at improving maternal and child nutrition. However, little is documented on maternal dietary practices, dietary diversity and the associated factors and composition of maternal foods in the study area. The objective of this study was therefore to assess the dietary practices, diversity, composition, and nutrient adequacy of diets of lactating mothers in Jimma zone, Southwest Ethiopia.

## Methods

### Setting and study design

The study was conducted in Southwest Ethiopia, Jimma zone. Three districts were purposively selected based on their agricultural production. Omo Nada, Dedo and Mana are cereal, vegetable and coffee producing areas, respectively. The survey was a community-based cross-sectional study.

### Participants, sample size and sampling technique

The study population included all lactating mothers in the study area. From these, 558 mothers were sampled following a stratified multistage sampling procedure [15]. On the first stage, the districts were stratified into lowland, midland and highland agroecology. Out of the 19 districts in the zone, three (1 Lowland, 1 Midland and 1 Highland) were selected purposively. The total kebeles (the smallest administrative unit of Ethiopia) in each district/woreda (the third level of the administrative division of Ethiopia—after zones and the regional states) were initially stratified into rural and urban areas. Then, three (two rural and one urban) were selected from the existing kebeles based on probability non-proportional to size sampling technique. A total of six rural and three urban kebeles were sampled from the three districts. In these selected kebeles, households with lactating mothers were systematically identified using the registration at the health posts (January 2014, post-harvest season which is dry). The calculated sample (558) was non-proportionally allocated to the selected districts (186 for each) then to the selected kebeles (62 for each kebele). A simple random sampling of the households was employed to recruit the study subjects. Sick mothers were excluded from the sample. Participants were recruited in February 2014.

### Variables

The variables involved in the analysis include socio-economic and demographic factors. These factors included family composition, age, household size, educational status of the mothers and fathers (formal vs. informal), the primary occupation of mothers and fathers, the wealth of the household and source of income, source of drinking water and availability of toilet facility, availability of health centres and education or training given on health and nutrition. Additionally, maternal dietary practices, maternal dietary diversity, nutrient compositions of maternal diets and nutrient adequacy of maternal diets were also involved in the analysis.

### Measurements

**Socio-economic information.**   A wealth index was determined using principal component analyses based on data on assets encompassing land for agriculture, production of crops, ownership of animals and properties [16]. The index was rank divided into tertiles and used for further analyses.

**Maternal practice of essential nutrition actions.**   The guideline on essential nutrition actions for healthy maternal nutrition during breastfeeding was followed to assess dietary practices of the mothers. Maternal knowledge and practice in use of insecticide-treated net, consumption of two additional meals in a day, consumption of a variety of food groups, daily use of iodized salt and family planning were assessed [17]. Semi-structured interviews were used to collect data on demographic, socio-economic and dietary practices.

**Dietary diversity.**   A single 24-h dietary recall was used to obtain data on dietary diversity. Dietary diversity was assessed with the ten MDD-W food groups namely (1) Grains, white roots and tubers, and plantains, (2) Pulses (beans, peas and lentils), (3) Nuts and seeds (4)

Dairy, (5) Meat, poultry and fish, (6) Eggs, (7) Dark green leafy vegetables, (8) Other vitamin A-rich fruits and vegetables, (9) Other vegetables and (10) Other fruits. One point was awarded for the consumption of each food group with a total possible score being 10. MDD-W was achieved if a mother consumed five or more food groups per day and respondents with mean of less than 5 food groups were considered as not achieving the MDD-W [18].

**Nutrient composition of maternal diets.**   Based on the responses on maternal dietary practices, twelve customary foods consumed by the mothers were selected for composition analysis. Composite samples (a mixture of individual samples) were collected, dried, ground and packaged [19]. These samples were analysed for proximate composition (protein, fat, carbohydrate, moisture, ash and fibre), energy content, mineral (iron, zinc, calcium and phosphorous) and anti-nutritional factors (phytate and tannin) following the respective official methods of analysis [20].

**Nutrient adequacy of maternal diets.**   According to Public Health Department of Georgia and World Health Organization, the average recommended portion per day is 550g/day, 450g/day and 175g/day for cereal, vegetable and legume food types, respectively for a healthy life [21]. Nutrient intake was calculated as a multiple of recommended portion per day and composition (proximate and mineral) of the food as indicated in the equation below.

$$Nutrient\ intake\left(\frac{g}{day}\right) = Recommended\ portion\left(\frac{g}{day}\right) * Nutrient\ composition\left(\frac{g}{100g}\right)\ (1)$$

Nutrient adequacy ratio (NAR) was calculated as the ratio of a subject's nutrient intake to the estimated average requirement calculated using the Food and Agriculture Organization/World Health Organization recommended nutrient intakes for mothers [3–5]. The mean adequacy ratio (MAR) was calculated as the sum of NARs for all evaluated nutrients divided by the number of assessed nutrients, expressed as a percentage [22] as indicated in the equations below.

$$NAR = \frac{Actual\ intake\ of\ a\ nutrient\ per\ day}{Recommended\ Daily\ Allowance\ of\ the\ nutrient}\ (2)$$

$$MAR = \frac{\sum NAR\ (each\ truncated\ at\ 1)}{Number\ of\ nutrients}\ (3)$$

## Statistical analysis

Data were entered, cleaned and then analyzed using SPSS version 20. Descriptive summaries using frequencies and proportions were used to present the study results. In this study, the dependent variable was dietary diversity coded one as lower dietary diversity and 0 as moderate or high dietary diversity. Descriptive statistics such as mean, median, range and percentages were calculated. Bivariate and multivariable logistic regression statistical analysis was carried out to determine the factors associated with diet diversity in lactating mothers. For all statistical analyses, a P value $< 0.05$ was considered for statistical significance.

## Ethics approval and consent to participate

Ethical clearance was obtained from the Research and Ethical Review Board of Jimma University. Permission to undertake the study was obtained from every relevant authority in Jimma Zone. Each study participant was briefed about the research and offered the opportunity to ask questions. Then oral informed consent was obtained from each participant before participation in the study, and data were kept confidential. The Ethical Review Board agreed to verbal

consent as the study was not intervention. The data collection and consent process were randomly checked by the Ethical Review Board to ensure the ethical undertaking of the research.

# Results

## Characteristics of the sample

A total of 558 mothers who had children aged 0–24 months were interviewed, and all agreed to participate in the study, which made the response rate of 100%. The majority 372 (66.7) of the participants were from the rural part of the survey area, Muslims in religion 516 (92.5%), Oromo in ethnicity 486 (87.1%), married and living together 501 (89.8%) and 336 (60.2%) were uneducated. A large proportion of the families belong to lower middle class or socio-economic status because most of them were engaged in small business, and many 436 (78.1%) were housewives (Table 1).

**Table 1. Maternal and household characteristics (Jimma Zone, Southwest Ethiopia 2014).**

| Characteristics | N | % |
|---|---|---|
| Maternal Age (Years) | | |
| 15–19 | 26 | 4.7 |
| 20–29 | 360 | 64.5 |
| 30–39 | 161 | 28.9 |
| 40–49 | 11 | 2 |
| Maternal education | | |
| Informal education | 338 | 60.6 |
| Formal education | 220 | 39.4 |
| Maternal Occupation | | |
| Housewife | 436 | 78.1 |
| Merchant | 83 | 14.9 |
| Other[$] | 39 | 7 |
| Maternal Marital status | | |
| Single | 33 | 5.9 |
| Married | 501 | 89.8 |
| Separated | 16 | 2.9 |
| Widowed | 5 | 0.9 |
| Divorced | 3 | 0.5 |
| Religion | | |
| Muslim | 516 | 92.5 |
| Orthodox | 33 | 5.9 |
| Protestant | 9 | 1.6 |
| Residence | | |
| Rural | 372 | 66.7 |
| Urban | 186 | 33.3 |
| Number of children per mother | | |
| 1–2 children | 252 | 45.2 |
| 3 and above children | 306 | 54.8 |
| District | | |
| Coffee producing | 186 | 33.3 |
| Cereal producing | 186 | 33.3 |
| Vegetable producing | 186 | 33.3 |
| Family Size | | |

*(Continued)*

**Table 1.** (Continued)

| Characteristics | N | % |
|---|---|---|
| 2–4 Members | 201 | 36 |
| 5–7 Members | 254 | 45.5 |
| 8–9 Members | 81 | 14.5 |
| 10 and Above Members | 22 | 3.9 |
| Socio-economic Status | | |
| Low | 186 | 33.3 |
| Medium | 189 | 33.9 |
| High | 183 | 32.8 |
| Husband's education | | |
| Informal education | 214 | 41.4 |
| Formal education | 303 | 58.6 |
| Husband's Occupation | | |
| Farmer | 318 | 63.5 |
| Daily Laborer | 63 | 12.6 |
| Merchant | 59 | 11.8 |
| Other[$] | 61 | 12.2 |
| Primary source of household income | | |
| Farming | 347 | 62.2 |
| Wage/Salary | 111 | 19.9 |
| Business | 90 | 16.1 |
| Salary | 44 | 7.9 |
| Other[¥] | 10 | 1.8 |

[$]Farmer, government employee, NGO employee, daily labourer

[$]Government Employee, NGO Employee, daily labourer.

[¥]Pension, remittance, commission.

## Maternal practice of essential nutrition actions

The majority of the lactating mothers reported that they received the following health information: have two additional meals a day (84.1%), consume a variety of food groups (82.8%), use insecticide-treated net (94.4%), and family planning (97.5%). On the contrary, only 39.1% of the participants reported receiving information to take iodized salt daily. In practice, only 22.2% of the respondents consume two additional meals per day, 14.5% reported consumption of a variety of food groups during lactation, and 36% reported use of iodized salt daily. Two-third of the mothers (65.6%) used bed net to protect themselves from malaria, and 61.5% of the mothers reported practising family planning (Table 2).

**Table 2. Maternal practice of essential nutrition actions during lactation in three districts of Jimma Zone, Southwest Ethiopia.**

| Variables | Educated | | Practised | |
|---|---|---|---|---|
| | n | % | n | % |
| Use insecticide-treated net | 527 | 94.4 | 366 | 65.6 |
| Consume two additional meals a day | 469 | 84.1 | 124 | 22.2 |
| Consume a variety of food groups | 462 | 82.8 | 81 | 14.5 |
| Daily use of iodized salt | 218 | 39.1 | 201 | 36.0 |
| Family planning | 544 | 97.5 | 343 | 61.5 |

**Table 3. Dietary diversity score and percentage of women consuming each of the 10 food groups.**

|  | DDS |
|---|---|
| Mean | 3.73 |
| Median | 4.0 |
| SD | 1.03 |
| Minimum | 0 |
| Maximum | 8 |
| MDD-W | N (%) |
| <5 food groups | 447 (80.1) |
| ≥5 food groups | 111 (19.9) |
| Consumption | N (%) |
| Grains, white roots and tubers, and plantains | 553 (99.1) |
| Pulses (beans, peas and lentils) | 431 (77.2) |
| Nuts and seeds | 13 (2.3) |
| Dairy | 75 (13.4) |
| Meat, poultry and Fish | 18 (3.2) |
| Eggs | 7 (1.3) |
| Dark green leafy vegetables | 299 (53.6) |
| Other vitamin A-rich vegetables and fruits | 58 (10.4) |
| Other vegetables | 499 (89.4) |
| Other fruits | 129 (23.1) |

## Dietary diversity

Table 3 summarises the DDS and percentage of women consuming each of the ten food groups. Among the ten food groups, the mean ± SD DDS of the study participants was 3. 73 ±1.03: with a maximum score of 8 and a minimum score of 0. The majority of the participants (80.1%) did not achieve MDD-W whereas 19.9% have achieved MDD. Results of the 24-hour recall showed that almost all of the mothers consume starchy staples (99.1%) and nearly three-fourth (77.2%) of the mothers consume pulses. Only 13.4% of the participants consumed dairy products in the 24 hours before the survey, and only 2.3% of them consumed nuts and seeds while only 3.2% consumed other meat, poultry and fish and only 1.3% consumed eggs. Half (53.6%) of the respondents reported consumption of dark green leafy vegetables. The use of vitamin A-rich fruits and vegetables was low (3.4%). However, most of the respondents (89.4%) reported consumption of other vegetables.

Table 4 shows the association between socio-demographic variables and maternal dietary diversity. Accordingly, dietary diversity score was significantly (P<0.01) affected by districts, residence area, ethnicity, husband's educational level and the socio-economic situation of the household. Oromo mothers who live in cereal producing districts, living in poor households and whose husbands attended lower-level education were more likely not to achieve MDD-W as compared with their respective counterparts.

Multivariable logistic regression analyses are presented in Table 5. After adjusting for different variables, meeting MDD-W was positively associated with agricultural production diversity(P = 0.001) and educational level of the women (P = 0.04). For a unit increase in agricultural production diversity of the household, the likelihood of achieving MDD-W was 1.4 times higher (AOR: 1.394, 95%CI: 1.142, 1.701). Similar, for one level increase in the educational status of women the odds of achieving MDD-W were 1.5 times higher (AOR: 1.485, 95%CI: 1.018, 2.168).

**Table 4. Distribution of maternal dietary diversity by different variables in Jimma Zone, Southwest Ethiopia.**

| Variables | Did not achieve MDD-W | Achieved MDD-W | |
|---|---|---|---|
| | n (%) | n (%) | P |
| Districts | | | |
| Coffee producing | 146 (78.5) | 40 (21.5) | 0.019* |
| Cereal producing | 161 (86.6) | 25 (13.4) | |
| Vegetable producing | 140 (75.3) | 46 (24.7) | |
| Ethnicity | | | |
| Oromo | 397 (81.7) | 89 (18.3) | 0.005* |
| Amhara | 16 (80.0) | 4 (20.0) | |
| Guraghe | 6 (75.0) | 2 (25.0) | |
| Yem | 6 (40.0) | 9 (60.0) | |
| Other | 22 (75.9) | 7 (24.1) | |
| Educational status of the woman | | | |
| 1 to 4 | 76 (78.4) | 21 (21.6) | 0.134 |
| 5 to 8 | 65 (76.5) | 20 (23.5) | |
| 9 to 10 | 19 (65.5) | 10 (34.5) | |
| Preparatory/TVET | 4 (44.4) | 5 (55.6) | |
| Diploma | 1 (50.0) | 1 (50.0) | |
| | | | |
| Husband's educational level | | | |
| 1 to 4 | 73 (83.9) | 14 (16.1) | 0.004* |
| 5 to 8 | 132 (78.6) | 36 (21.4) | |
| 9 to 10 | 26 (57.8) | 19 (42.2) | |
| Preparatory/TVET | 17 (89.5) | 2 (10.5) | |
| Diploma | 2 (50) | 2 (50) | |
| Place of residence | | | |
| Rural | 297 (79.8) | 75 (20.2) | 0.822 |
| Urban | 150 (80.6) | 36 (19.4) | |
| Socio-economic status | | | |
| Poor | 155 (83.3) | 31 (16.7) | 0.046* |
| Medium | 154 (82.8) | 32 (17.2) | |
| Rich | 138 (74.2) | 111 (19.9) | |

* = Significant at the 0.05 level.

Conversely, district of the study (P = 0.003) and place of residence (P = 0.019) were negatively associated with meeting MDD-W. Women living in cereal producing (Omo Nada) District were 86% less likely to meet the MDD-W as compared to those who live in vegetable producing (Dedo) District (AOR: 0.236, 95%CI" 0.090, 0.616). Likewise, women who live in the rural areas were 72% less likely to meet the MDD-W as compared to those who live in the urban areas (AOR: 0.283, 95%CI: 0.099, 0.810).

## Nutrient composition of maternal diets

The proximate composition and calorific value of the sampled maternal foods ranged between 24.8–65.6g/100g for moisture, 7.6–19.8 g/100g for protein, 2.1–23.1 g/100g for crude fat, 2.0–27 g/100g crude fibre, 1.0–21.2 g/100g for total ash, 0.9–45.8 g/100g for total carbohydrate and 124.5–299.6Kcal/100g for energy content (Table 6). The mineral contents ranged between 9.5–52.5mg/100g for iron, 2.2–4.2mg/100g for zinc, 42.6–318.2mg/100g for calcium, and

**Table 5. Multivariable logistic regression model predicting the likelihood of having MDD-W in Jimma Zone, Southwest Ethiopia.**

| Model | B | P | AOR | 95% C.I. | |
|---|---|---|---|---|---|
| | | | | Lower | Upper |
| District | | | | | |
| Coffee producing (Mana) | -0.211 | 0.599 | 0.810 | 0.369 | 1.779 |
| Cereal producing (Omo Nada) | -1.444 | 0.003 | 0.236 | 0.090 | 0.616 |
| Vegetable producing (Dedo) | | | 1.000 | | |
| Food production diversity | 0.332 | 0.001 | 1.394 | 1.142 | 1.701 |
| Educational level of the women | 0.396 | 0.040 | 1.485 | 1.018 | 2.168 |
| Wealth tertile | | | | | |
| Poor | -0.732 | 0.154 | 0.481 | 0.176 | 1.317 |
| Medium | -1.013 | 0.062 | 0.363 | 0.125 | 1.052 |
| Rich | | | 1.000 | | |
| Age of the woman (yrs) | -0.065 | 0.135 | 0.937 | 0.861 | 1.020 |
| Household has alternative income source | -0.175 | 0.639 | 0.839 | 0.404 | 1.744 |
| Place of residence, | | | | | |
| Rural | -1.263 | 0.019 | 0.283 | 0.099 | 0.810 |
| Urban | | | 1.00 | | |
| Educational status of the husband | 0.894 | 0.085 | 2.446 | 0.883 | 6.776 |
| Household size | 0.150 | 0.171 | 1.162 | 0.937 | 1.441 |

Production diversity: a score generated by summing the number of animal and plant source food that the household reported to produce.

Maximum Standard error: 0.543, Hosmer Lemeshow Test for Model Fitness (P = 0.052).

AOR: Adjusted Odds Ratio.

CI: Confidence Interval.

**Table 6. The proximate and energy composition (dry weight basis) of sampled maternal foods in three districts of Jimma Zone, Southwest Ethiopia from March-May 2014.**

| Food types | Protein | Fat | Carbohydrate | Fibre | Ash | Energy |
|---|---|---|---|---|---|---|
| | g/100g | g/100g | g/100g | g/100g | g/100g | Kcal/100g |
| Lentil sauce | 18.2 | 23.1 | 4.8 | 5.2 | 11.9 | 299.6 |
| Bean sauce | 19.8 | 17.5 | 10 | 5.3 | 11 | 276.5 |
| Pea powder sauce | 16.8 | 19. 8 | 2.2 | 4.3 | 16.9 | 254.2 |
| Kale-bean sauce | 19.6 | 6.5 | 0.9 | 27 | 21.2 | 140.5 |
| *Injera* (T) | 10.1 | 2.6 | 45.6 | 2.5 | 1.5 | 246 |
| *Injera* (T+S) | 10.7 | 2.8 | 43.8 | 2.2 | 1.8 | 243.5 |
| *Injera* (T+M) | 9.6 | 7.5 | 34.8 | 7.6 | 2.6 | 245.1 |
| *Injera* (T+M+S) | 9 | 3.3 | 39.2 | 12.2 | 2 | 222.3 |
| *Injera* (T+S+R) | 9.9 | 5.5 | 8.9 | 8 | 2.1 | 124.5 |
| Unleavened bread (W+M) | 10.8 | 2.1 | 45.8 | 2 | 1.2 | 245.2 |
| Unleavened bread (M) | 7.6 | 3.5 | 42.8 | 2 | 1 | 233.1 |
| Unleavened bread (W) | 11.5 | 3.5 | 33.3 | 2.1 | 1.5 | 210.5 |
| RDA (g/day) [$] | 65 | 69.1 | 210 | 21–25 | NA | 2750[$] |

T = Teff; S = Sorghum; M = Maize; R = Rice; W = Wheat; RDA = Recommended Daily Allowance.

[$][4]

[$][23]; Unit for energy it is Kcal/day.

**Table 7. The content of minerals and anti-nutritional factors (dry weight basis) of sampled maternal foods in three districts of Jimma Zone, Southwest Ethiopia.**

| Food types | Iron | Zinc | Calcium | Phosphorous | Phytate | Tannin |
|---|---|---|---|---|---|---|
| | mg/100g | mg/100g | mg/100g | mg/100g | mg/100g | mg/100g |
| Lentil sauce | 20 | 3.4 | 0.2 | 0.3 | 92 | 14.17 |
| Bean sauce | 20.5 | 3.4 | 0.2 | 0.4 | BDL | 27.06 |
| Pea powder sauce | 27.6 | 4.2 | 0.3 | 0.3 | 11.06 | 65.26 |
| Kale-bean sauce | 24.8 | 2.8 | 0.1 | 0.3 | 47.65 | 205.05 |
| *Injera* (T) | 21.8 | 3.4 | 0.3 | 0.3 | 120.45 | BDL |
| *Injera* (T+S) | 39.7 | 3.2 | 0.2 | 0.4 | 145.01 | 81.64 |
| *Injera* (T+M) | 52.5 | 3.1 | 0.3 | 0.3 | 178.89 | 6.64 |
| *Injera* (T+M+S) | 39.4 | 2.7 | 0.2 | 0.4 | 121.61 | 315.85 |
| *Injera* (T+S+R) | 22 | 2.2 | 0.2 | 0.2 | 150.82 | 13.71 |
| Unleavened bread (W+M) | 9.5 | 2.2 | 0.1 | 0.3 | 80.21 | 3.71 |
| Unleavened bread (M) | 12.5 | 2.5 | 0.3 | 0.3 | BDL | 8.36 |
| Unleavened bread (W) | 20.7 | 3.8 | 0.2 | 0.3 | 55.17 | 5.46 |
| RDA (mg/day) | 10 | 12 | 1000 | 1000 | NA | NA |

T = Teff; S = Sorghum; M = Maize; R = Rice; W = Wheat; BDL = Below Detectable Level.

150.7–379.9mg/100g for phosphorus. The content of anti-nutritional factors ranged between 11.1–178.9mg/100g for phytate and 3.7–315.9mg/100g for tannin (Table 7).

## Nutrient adequacy of maternal diets for achieving maternal nutritional goals

Out of the twelve foods assessed in this study, most of the maternal diets do not provide the recommended daily allowances for protein, fat and energy (NAR<1). Most of the cereal-based foods contain sufficient minerals (NAR>1). The overall adequacy is less than 1 for all diets (MAR<1) (Table 8) which indicates the requirements for all the nutrients were not met.

**Table 8. Adequacy of the maternal foods in nutrients for meeting dietary recommendations.**

| Food types | P | F | CHO | Fi | Energy | Fe | Zn | Ca | P | MAR |
|---|---|---|---|---|---|---|---|---|---|---|
| Lentil sauce | 0.5 | 0.6 | 0.0 | 0.3 | 0.2 | 3.5 | 0.5 | 0.4 | 0.5 | 0.44 |
| Bean sauce | 0.5 | 0.4 | 0.1 | 0.3 | 0.2 | 3.6 | 0.5 | 0.4 | 0.7 | 0.45 |
| Pea powder sauce | 0.5 | 0.5 | 0.0 | 0.3 | 0.2 | 4.8 | 0.6 | 0.5 | 0.5 | 0.45 |
| Kale-bean sauce | 1.4 | 0.4 | 0.0 | 4.1 | 0.2 | 11.2 | 1.1 | 0.2 | 1.3 | 0.65 |
| *Injera* (T) | 0.9 | 0.2 | 1.2 | 0.5 | 0.5 | 12.0 | 1.5 | 1.8 | 1.9 | 0.78 |
| *Injera* (T+S) | 0.9 | 0.2 | 1.2 | 0.4 | 0.5 | 21.9 | 1.5 | 1.3 | 2.1 | 0.78 |
| *Injera* (T+M) | 0.8 | 0.6 | 0.9 | 1.4 | 0.5 | 28.9 | 1.4 | 1.7 | 1.8 | 0.86 |
| *Injera* (T+M+S) | 0.8 | 0.3 | 1.0 | 2.2 | 0.4 | 21.7 | 1.2 | 1.0 | 2.1 | 0.82 |
| *Injera* (T+S+R) | 0.8 | 0.4 | 0.2 | 1.5 | 0.3 | 12.1 | 1.0 | 1.0 | 0.8 | 0.73 |
| Unleavened bread (W+M) | 0.9 | 0.2 | 1.2 | 0.4 | 0.5 | 5.2 | 1.0 | 0.5 | 1.5 | 0.71 |
| Unleavened bread (M) | 0.6 | 0.3 | 1.1 | 0.4 | 0.5 | 6.9 | 1.2 | 1.5 | 1.5 | 0.75 |
| Unleavened bread (W) | 1.0 | 0.3 | 0.9 | 0.4 | 0.4 | 11.4 | 1.7 | 0.8 | 1.9 | 0.75 |

P = Protein; Fat = Fat; CHO = Carbohydrate; Fi = Fibre; Cal = Energy Content; Fe = Iron; Zn = Zinc; Ca = Calcium; P = Phosphorous. T = Teff; S = Sorghum; M = Maize; R = Rice; W = Wheat.

## Discussion

### Maternal practice of essential nutrition actions

The majority of the study participants did not change their previous food intake habit (quantity, quality and diversity) during lactation, which is in contrast to the recommendations in the essential nutrition actions to improve maternal health and nutrition [17]. It is known that a lactating mother should produce about 700 to 800ml of milk per day and this requires an additional energy need of about 550 calories (two extra meals) per day [2, 4, 24]. Our finding on the practice of consumption of iodized salt daily (36%) was lower than 76.3% reported in the capital city, Addis Ababa [25]. This might be due to the lesser accessibility of different media (FM radios and television) nearby for the target group in the study area.

### Dietary diversity

The women's DDS was similar to those previously reported in Bangladeshi women as well as in women in Burkina Faso, Mali, Mozambique, Bangladesh, and the Philippines using the same definition of dietary diversity [26]. Mothers who live in the cereal producing district and rural villages, who are in 25–29 years age group, illiterate and poor mothers had lower DDS as compared with their respective counterparts. These results are in agreement with previous reports which indicated that there is a positive relationship between dietary diversity and diversified agricultural production and income from the agricultural product [27], and formal education to the mother [28]. Regarding the association between maternal age and DDS, a contradicting report has been documented from Nigeria where being young age was positively associated with dietary diversity [24]. In the coffee producing district, mothers could have enough money for household expenditure to buy various food commodities. They might have a diverse diet compared to mothers living in cereal producing regions. Therefore, the study participants highly rely on a few staple foods and often include little or no animal products and few fresh fruits and vegetables as compared to Dedo and Mana districts. This finding is consistent with previous reports [27–29] who stated that diversified agricultural production and income from agricultural product market even in a more impoverished household has a positive relationship with dietary diversity. The variation in dietary diversity between vegetable and cereal producing districts indicates that households tend to consume what they have produced; perishables (vegetables) are more likely to be consumed by the producers than durables (cereals) as they spoil fast if not sold. Therefore, increasing the diversity of farm should be encouraged as one key strategy to improve diet diversity and quality [30].

Similarly, DDS was significantly different among lactating mothers living in urban and rural residences. A large proportion (59.1%) of mothers in the rural area showed a low DDS than an urban residential area. Previous findings by Ruel (2003) also indicated urban households had a consistently higher dietary diversity than rural households. Considering the fact that rural areas are hubs for agricultural production, one can assume rural women could be better in dietary diversity than urban counterparts. However, production diversification may not always mean dietary diversity. For example, a study in rural Nigeria found out that production diversification has no statistically significant effect on the dietary diversity of households [30]. Other researchers also argue that despite theoretical basis for the correlation between production diversity and dietary diversity, "there is a need for a deeper empirical understanding of how, under what circumstances, and through what pathways own-production of nutritious foods improves diets [31]". Although in contrast, higher DDS was reported in the rural sector [28].

According to Ajani, educational status and household wealth were interrelated and had a positive association with dietary diversity [28]. Higher education attainment is likely to be

## Composition of maternal diets

Considering the average recommended gram of portion per day, 550g/day, 450g/day and 175g/day for cereal, vegetable and legume food types, respectively, only kale sauce can fulfil the RDA for protein (17.33g/day) [4, 23]. Kale sauce is prepared by cooking onion and oil 4 to 5 minutes and adding chopped and cooked-until-soft kale, and stirring once or twice, for 10 minutes. During collection of food samples, we have observed that most households add protein rich ingredients like beans when making kale sauce, and the name kale sauce could be a misnomer. Only unleavened bread made of a blend of wheat and corn and *Injera* (Ethiopian traditional cereal-based pancake, prepared from fermented cereal sour dough) made of teff can fulfil the RDA for carbohydrate (250 g/day) [4]. Concerning the fibre content, four out of twelve food types fulfil the RDA. These are kale sauce, *Injera* made of teff, maize and sorghum, *Injera* made of teff, maize and rice and, *Injera* made of teff and maize [4, 23]. Even though there is no recommendation of ash for lactating mothers, it is known that as ash content (a measure of the total amount of minerals present within a food) is correlated with micronutrient content in that particular food type. Therefore, from the 12 food types analysed, kale sauce could provide relatively healthy micronutrients to the lactating mothers. Based on the recommended gram of portion for adults none of the 12 food types fulfils the RDA for fat (69.1 g/day) [4], and energy (2750 kcal/day) [23].

Among the mineral forms analysed, iron was predominantly found in all tested foods above the recommended amount (10mg/day). Except for lentil, bean and pea powder sauces, all the food types can fulfil the recommended daily allowance for zinc (12mg/day). All *Injera* types and the unleavened bread made of maize can fulfil the RDA for calcium (Ca). Except for lentil, bean and pea powder sauces and *Injera* made of blends of teff, sorghum, and rice, all the food types can fulfil the recommended daily allowance for P (1000mg/day) [4].

High phytate intakes and low fruit consumption may compromise iron, zinc, and to a lesser extent calcium status in these mothers [32]. Tannins had been reported to affect protein digestibility, adversely influencing the bioavailability of non-haem iron leading to poor iron and calcium absorption [33]. There is no recommendation or cut-off points for the availability of anti-nutritional factors in foods while reducing the anti-nutrition levels (e.g. dephytinization) to the best minimum using different strategies is advised [34] combined with enrichment with animal-source foods and/or fortification with appropriate levels and forms of mineral fortificants [35].

## Nutrient adequacy of maternal diets for achieving maternal nutritional goals

The ideal cut-off for nutrient adequacy should be 1, which would mean that all the nutrients were consumed in a sufficient amount [36]. In another way, the NAR for protein and calorific value/energy throughout the whole sampled food types were found to be below 1, which means that all the commonly eaten foods by lactating mothers were not sufficient to meet the protein and energy requirements. Protein from cereals is usually with relatively low digestibility and quality [37]. The consequences of low energy in the diets is the women suffer from chronic energy deficiency and will have poor nutritional status [2]. Lack of dietary diversification and the foods lack vegetables, fruits, and animal source food types [29]. The overall nutrient adequacy was different for the twelve types of food types. MAR was below the cut-off point for all food types.

As presented in Table 8, most of the food types had sufficient mineral content which had NAR above 1. These could be, from the supply side, because of Ethiopian foods, especially *Injera*, had a significantly high amount of minerals [2]. The other reason, from a demand side, could be since breastfeeding usually suppresses menstruation for a few months, lactating mothers iron requirement is decreased to 10 mg/day RDA [36]. Care should be taken in interpreting these results as bioaccessibility and bioavailability of minerals from plant-based foods is impaired [38].

## Limitations and strengths of the study

**Limitation of the study.** Recall bias might have occurred as information on maternal dietary practices were collected retrospectively. The 24-hour recall is prone to over and under-reporting of maternal dietary intakes. Dietary intakes of the mothers were not directly measured by using portion sizes as family members in Ethiopia consume food from a common plate. Ethiopia does not have a food guide; therefore we used the amount of recommended portions per day (in grams) for Georgia. Vitamin contents of the meals were not analysed. Ethiopia is a very diverse country in terms of food production and consumption culture, and this study may not be representative of the rest of the country. Additionally, the dietary data alone does not refer to bioavailability of the nutrients, which are influenced by the overall health of the mothers.

**Strength of the study.** Rigorous sampling methodology was used. An official food analysis methodology was employed to analyze the composition of undocumented maternal diets.

## Conclusions

Our study shows that lactating women in Southwest Ethiopia consume diets that lack sufficient nutrient-dense foods to achieve micronutrient adequacy. The dietary diversity of lactating mothers in the study area was not satisfactory. Kale sauce, which is traditionally popular in South and Southwest part of rural Ethiopia, tend to provide relatively healthy macro and micronutrients to the lactating mothers. A community-based nutritional education on food diversification and local processing techniques based on a multi-sectoral approach is needed to curb the problem of malnutrition among lactating mothers in the study area.

## Supporting information

**S1 Checklist. STROBE checklist.**
(PDF)

**S1 Data. Minimal underlying data set on maternal dietary practices, dietary diversity, and nutrient composition of diets of lactating mothers in Jimma zone, Southwest Ethiopia.**
(DOCX)

**S2 Data. Type of stews consumed and ingredients of the stews.**
(SPV)

## Acknowledgments

We are grateful to the mothers for their participation and the health extension workers for their assistance. The authors also thank Ethiopian Public Health Institute for assisting in the nutrient analysis.

## Author Contributions

**Conceptualization:** Sirawdink Fikreyesus Forsido, Tefera Belachew, Oliver Hensel.

**Data curation:** Sirawdink Fikreyesus Forsido, Frehiwot Tadesse, Tefera Belachew.

**Formal analysis:** Sirawdink Fikreyesus Forsido, Frehiwot Tadesse, Tefera Belachew.

**Funding acquisition:** Oliver Hensel.

**Investigation:** Frehiwot Tadesse.

**Methodology:** Sirawdink Fikreyesus Forsido, Tefera Belachew.

**Project administration:** Oliver Hensel.

**Resources:** Tefera Belachew, Oliver Hensel.

**Validation:** Tefera Belachew, Oliver Hensel.

**Writing – original draft:** Sirawdink Fikreyesus Forsido, Frehiwot Tadesse.

**Writing – review & editing:** Sirawdink Fikreyesus Forsido, Tefera Belachew, Oliver Hensel.

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
