## [Decision Letter · Decision Letter 0]

12 Feb 2021

PONE-D-20-19141

Maternal dietary practices, dietary diversity, and nutrient composition of Diets of Lactating Mothers in Jimma Zone, Southwest Ethiopia

PLOS ONE

Dear Dr. Forsido,

Thank you for submitting your manuscript to PLOS ONE. After careful consideration, we feel that it has merit but does not fully meet PLOS ONE’s publication criteria as it currently stands. Therefore, we invite you to submit a revised version of the manuscript that addresses the points raised during the review process.

See the comments by the reviewers.

We look forward to receiving your revised manuscript.

Kind regards,

Gideon Kruseman, Ph.D.

Academic Editor

PLOS ONE

Journal Requirements:

2.We note that the grant information you provided in the ‘Funding Information’ and ‘Financial Disclosure’ sections do not match.

3.In your Data Availability statement, you have not specified where the minimal data set underlying the results described in your manuscript can be found. PLOS defines a study's minimal data set as the underlying data used to reach the conclusions drawn in the manuscript and any additional data required to replicate the reported study findings in their entirety. All PLOS journals require that the minimal data set be made fully available. For more information about our data policy, please see http://journals.plos.org/plosone/s/data-availability.

4. We noted in your submission details that a portion of your manuscript may have been presented or published elsewhere.

"This work has been presented at a conference Tropentag 2016 (https://www.tropentag.de/2016/abstracts/full/100.pdf)."

Reviewers' comments:

Reviewer's Responses to Questions

**Comments to the Author**

1. Is the manuscript technically sound, and do the data support the conclusions?

Reviewer #1: Yes

Reviewer #2: Yes

Reviewer #3: No

Reviewer #4: Partly

2. Has the statistical analysis been performed appropriately and rigorously? 

Reviewer #1: Yes

Reviewer #2: Yes

Reviewer #3: No

Reviewer #4: No

3. Have the authors made all data underlying the findings in their manuscript fully available?

Reviewer #1: Yes

Reviewer #2: No

Reviewer #3: No

Reviewer #4: No

4. Is the manuscript presented in an intelligible fashion and written in standard English?

Reviewer #1: Yes

Reviewer #2: No

Reviewer #3: No

Reviewer #4: No

5. Review Comments to the Author

Reviewer #1: Concerns that the data was collected about 6 years ago ie 2014

There are a few grammatical errors that should be reviewed in the manuscript and corrected - need to improve in certain parts on the English

There is reference in the abstract to "little stores" - I propose this be altered to minimal stores

Would have been good to make reference to the overall maternal mortality rate in Ethiopia including causes - and if any are related to nutritional deficiencies . More data on the general health of the subjects in the study would help

In mentioning the vulnerability of mothers to undernutrition in this context, would help to also mention the possibility of intergenerational cycles of malnutrition

Whilst most of the study participants were Muslim, I wondered the extent to which those who were christian were Orthodox Christians and could have still adhered to fasting regimes even during lactation - a comment on this aspect would be useful

In order to understand the food security, income and poverty dimension, it would help to know how many of the mothers resided in PSNP woredas/ kebelles and therefore may have been enrolled in this social protection scheme which could have influenced dietary diversity.

Whilst there is mention of consumption of iodised salt - at the time the survey data was collected the quality of iodised salt was very poor - with minimal levels

Under limitations I propose to include the fact that the dietary data alone does not refer to bioavailabilty of the nutrients which are influenced by the overall health of the mothers

Reviewer #2: Thank you for the opportunity to review this paper. My comments are as follows:

There are some minor issues with English throughout that need to be addressed. For example:

Line 169 "Majority" should be "The majority.."

Line 172 "When it comes to the practice.." should be "In practice.."

Line 210 "played its role for this" should be "played a role in this"

Line 214 "Majority" should be "The majority.."

Abstract: Line 17 "little stores" should perhaps be "low stores"

Line 27 DDS should be defined

Line 40 should be "nutrition education program"

Abstract is otherwise appropriate.

Introduction: Appropriate.

Methods: Maternal dietary practices section, line 109, what is meant by semi-structured questionnaires? Questionnaires are generally structured in nature - do the authors mean semi-structured interviews?

Line 115 to calculate DDS was 1 point awarded for the consumption of each food group with a total possible score being 9? This should be explained in the text.

Line 116-117 "greater than 3.4 mean food groups" This word is not correct and should be something like "a mean (or average) number of food groups greater than 3.4".

Line 117 "respondents with <3.4 mean food groups" is also not correct and should be "respondents with mean of less than 3.4 food groups..."

Line 121 Please clarify as to what composite samples are. Are they combined samples or just samples of each food individually?

Line 128 "gram of portion per day" is not clear. Should is just be "grams per day"?

Were t-test considered to compare means between groups?

Results:

Line 185 What is "organ" meat?

Line 190-191 sentence beginning "This study.." is not presenting results and should be removed.

Discussion: Line 215-216 is not clear and needs to be reworked.

Line 222 is not clear.

Line 232 could the authors please define what they mean by cash crop

Line 242, incorrect reference format for Ruel

Paragraph 240-244 is not clear e.g. what is "placed in the separate residence"?

Line 245 incorrect reference format for Ajani. Again, this paragraph is not clear.

Line 253, could the authors please describe what injera is. For example, is it like bread?

Line 257 please indicate what is meant by ash.

Line 266 has Ca been previously defined (it is assumed this is calcium?)

Line 251 - kale sauce has been highlighted specifically in the discussion but has not been described previously. It would be of benefit to describe how kale sauce is used, or what part it plays in the diet prior to this.

Limitations - appropriate although how representative is this study of the rest of Ethiopia? Strengths - I am unclear how this is a strength. This is an outcome from the study but the methodology undertaken to achieve this is the strength.

Conclusions:

Line 307-308 this sentence is not clear and should be reworked.

Table 4 - why was 0.01 used as the p value when .05 is the standard and this is stated in line 147.

Other tables appropriate although only descriptive in nature

Reviewer #3: While the topic of the study has public health importance, masurement approaches, the tools and the data analysis is not scientifically sounding. Despite the result section and conclusion attempted to note the variables associated with low dietary diversity there is no a strong method of data analysis to test the hypothesis. The study used simple chi-square which simply allows us to see the precence or absence of any association between two variables at a time. This does not help us to know which variable is cause and which is an effect? In what magnitude? In what direction (favoring or not).

Another concern is, construction of survey questionnaire has not been clearly stated and its reliability and validity are not presented. For example, measuring socioeconomic class, educational status etc was not clear.

Including different agroecological areas in this study could be taken as one of the strengths. However, it is true that dietary diversity could vary seasonally, and the study has not mentioned the season(s) when the study was undertaken.

Reviewer #4: Consideration of the nutrient needs of lactating women is an important topic, particularly as longer periods of lactation are being recommended.

An important reference that the authors will want to consider is: FAO and FHI 360. 2016. Minimum Dietary Diversity for Women: a Guide for Measurement. Rome: FAO. The MDD-W has been utilized and validated by several groups. The MDD-W presented in this publication outlines 10 food groups (which are not the same as those specified in this manuscript. And it specifies a cut-off of ≥5 food groups.

There are several inconsistencies in the manuscript that the authors must address. Some, but not all, of these inconsistencies are included in the following points:

L. 22-23 says the average MDD-W was computed from two 24-h recalls, while L. 112 says a single 24-h dietary recall….

The nine food groups listed on L. 113 -115 are not the same as the FAO publication nor are they identical to the food groups listed in Table 3 of this manuscript.

L. 88-89 says kebeles were sampled from the three selected districts by lottery method. If only by lottery, there could have been 0 to 3 urban kebeles selected from one district. Please clarify the sampling procedure.

L. 127. The document from the Public Health Department of Georgia and World Health Organization (2015) seems to be a report for the general public about healthy eating. Are the recommended grams of portion per day given as dry weight or wet weight. How were these portion sizes adjusted for the calculations in this paper?

L. 135 mentions the FAO/WHO RNI intakes for mothers. L. 139 specifies division by the RDA. The RNI and the RDA typically are not the same. RDA appears also on Table 5&6.

L. 180. Does the Dietary score of zero represent a woman who was fasting??

L. 210 Fermentation is very likely to increase the bioavailability of Zn, Fe, and perhaps Ca, but usual laboratory analyses and food composition tables show total mineral concentration not bioavailable mineral concentrations.

L. 222. What is meant by different media?

While it is clear that diet diversity was poor and food intakes were inadequate, it is not always clear which differences among the three areas were significantly different from each other. On Table 4, overall differences are shown between “low” and “moderate or high” by chi-square. In L. 224-235 which of the differences described are significantly different when there are more than two categories for a variable?

A supplementary table is needed to describe the ingredients (recipe) in lentil sauce, bean sauce, pea powder sauce and kale sauce. Particularly for a sauce based on kale, it is hard to imagine a protein concentration of 19.6 g/100 g.

There has been work published on phytate and bioavailability (Food & Nutrition Bulletin, 2010) that the authors might find useful.

6. PLOS authors have the option to publish the peer review history of their article (what does this mean?). If published, this will include your full peer review and any attached files.

Reviewer #1: No

Reviewer #2: No

Reviewer #3: No

Reviewer #4: No

---

## [Author Response · Author response to Decision Letter 0]

7 Apr 2021

Reviewers' comments:

Reviewer's Responses to Questions

Comments to the Author

1. Is the manuscript technically sound, and do the data support the conclusions?

Reviewer #1: Yes

Reviewer #2: Yes

Reviewer #3: No

Reviewer #4: Partly

2. Has the statistical analysis been performed appropriately and rigorously?

Reviewer #1: Yes

Reviewer #2: Yes

Reviewer #3: No

Reviewer #4: No

3. Have the authors made all data underlying the findings in their manuscript fully available?

Reviewer #1: Yes

Reviewer #2: No

Reviewer #3: No

Reviewer #4: No

4. Is the manuscript presented in an intelligible fashion and written in standard English?

Reviewer #1: Yes

Reviewer #2: No

Reviewer #3: No

Reviewer #4: No

5. Review Comments to the Author

Reviewer #1: 

Reviewer 1 comment 1:

Concerns that the data was collected about 6 years ago ie 2014

Authors' response 1:

That is true.

Reviewer 1 comment 2:

There are a few grammatical errors that should be reviewed in the manuscript and corrected - need to improve in certain parts on the English.

Authors' response 2:

Language edition is done using a premium subscription of Grammarly (a digital writing assistance tool based on artificial intelligence and natural language processing). See revised manuscript.

Reviewer 1 comment 3:

There is reference in the abstract to "little stores" - I propose this be altered to minimal stores

Authors' response 3:

Line 24: Corrected as suggested by the reviewer.

Reviewer 1 comment 4:

Would have been good to make reference to the overall maternal mortality rate in Ethiopia including causes - and if any are related to nutritional deficiencies. More data on the general health of the subjects in the study would help

Authors' response 4:

Lines 68-70: The following sentence is added to show the relation between nutritional deficiencies and maternal mortality.

“Ethiopia experiences one of the highest incidences of child and maternal nutritional deficiencies which contribute to increased morbidity and mortality (Central Statistical Agency [Ethiopia] & ICF International, 2012).”

Reviewer 1 comment 5:

In mentioning the vulnerability of mothers to undernutrition in this context, would help to also mention the possibility of intergenerational cycles of malnutrition

Authors' response 5:

Lines 70-72: The following sentence is added:

“A malnourished mother will give birth to a low birth weight baby; the low birth weight baby will grow as a malnourished child, then to a malnourished teenager, then to a malnourished pregnant woman, and so the cycle continues (10).”

Lines 471-473: The following reference was added to the reference list:

“Ramakrishnan U, Martorell R, Schroeder DG, Flores R. Role of intergenerational effects on linear growth. In: Journal of Nutrition [Internet]. American Institute of Nutrition; 1999 [cited 2021 Mar 25]. Available from: https://pubmed.ncbi.nlm.nih.gov/10064328/”

Reviewer 1 comment 6:

Whilst most of the study participants were Muslim, I wondered the extent to which those who were christian were Orthodox Christians and could have still adhered to fasting regimes even during lactation - a comment on this aspect would be useful

Authors' response 6:

Out of 558 participants, there were 42 Christian participants in the study out of which 33 (78.5%) were Orthodox Christians. However, there was neither fasting nor feasting when we undergone the survey.

Reviewer 1 comment 7:

In order to understand the food security, income and poverty dimension, it would help to know how many of the mothers resided in PSNP woredas/ kebelles and therefore may have been enrolled in this social protection scheme which could have influenced dietary diversity.

Authors' response 7:

Only chronically food insecure districts (woredas) are enrolled in PSNP. None of our study districts were in this social protection scheme.

Reviewer 1 comment 8:

Whilst there is mention of consumption of iodised salt - at the time the survey data was collected the quality of iodised salt was very poor - with minimal levels

Authors' response 8:

What we meant by the sentences in lines 312-315 is the consumption of iodized salt in the study area is very low compared to other areas of the country where media coverage (radio and television) coverage is wider.

Reviewer 1 comment 9:

Under limitations I propose to include the fact that the dietary data alone does not refer to bioavailabilty of the nutrients which are influenced by the overall health of the mothers

Authors' response 9:

Lines 400-402: The following sentence was added as suggested by the reviewer:

“Additionally, the dietary data alone does not refer to bioavailability of the nutrients, which are influenced by the overall health of the mothers.”

Reviewer #2: 

Thank you for the opportunity to review this paper. My comments are as follows:

Reviewer 2 comment 1:

There are some minor issues with English throughout that need to be addressed. For example:

Line 169 "Majority" should be "The majority.."

Authors' response 1:

Line 191: “Majority…” is changed to “The majority…”

Line 200: “Majority…” is changed to “The majority…”

The English was edited by a premium version of Grammarly.

Reviewer 2 comment 2:

Line 172 "When it comes to the practice.." should be "In practice.."

Authors' response 2:

Lines 203-204: Modified as suggested by the reviewer

Reviewer 2 comment 3:

Line 210 "played its role for this" should be "played a role in this"

Authors' response 3:

Line 297: The phrase “fermentation is believed to have played its role for this” is deleted as it is out of context.

Reviewer 2 comment 4:

Line 214 "Majority" should be "The majority.."

Authors' response 4:

Line 306: “Majority…” is changed to “The majority…”

Reviewer 2 comment 5:

Abstract: Line 17 "little stores" should perhaps be "low stores"

Authors' response 5:

Line 24: "little stores" is changed to "minimal stores"

Reviewer 2 comment 6:

Line 27 DDS should be defined

Authors' response 6:

Line 36: “DDS” was changed to “dietary diversity score (DDS)”

Reviewer 2 comment 7:

Line 40 should be "nutrition education program"

Authors' response 7:

Line 53: Corrected as suggested by the reviewer

Reviewer 2 comment 8:

Abstract is otherwise appropriate.

Authors' response 8:

Ok

Reviewer 2 comment 9:

Introduction: Appropriate.

Authors' response 9:

Ok

Reviewer 2 comment 10:

Methods: Maternal dietary practices section, line 109, what is meant by semi-structured questionnaires? Questionnaires are generally structured in nature - do the authors mean semi-structured interviews?

Authors' response 10:

Line 130: “questionnaires” is changed to “interviews”

Reviewer 2 comment 11:

Line 115 to calculate DDS was 1 point awarded for the consumption of each food group with a total possible score being 9? This should be explained in the text.

Authors' response 11:

Lines 142 - 143: The following sentence is added as suggested by the reviewer:

“One point was awarded for the consumption of each food group with a total possible score being 10.”

NB. As suggested by another reviewer, the FAO & FHI 360, 2016 methodology, which is based on 10 food groups, was used to recalculate the dietary diversity of the women.

Reviewer 2 comment 12:

Line 116-117 "greater than 3.4 mean food groups" This word is not correct and should be something like "a mean (or average) number of food groups greater than 3.4".

Authors' response 12:

Line 144-: corrected as suggested by the reviewer:

“mean number of food groups greater than 5”

NB. According to the FAO & FHI 360, 2016 methodology, which is based on 10 food groups, the cut-off is >5 food groups.

Reviewer 2 comment 13:

Line 117 "respondents with <3.4 mean food groups" is also not correct and should be "respondents with mean of less than 3.4 food groups..."

Authors' response 13:

Line 145: corrected as suggested by the reviewer:

“respondents with mean of less than 3.4 food groups”

Reviewer 2 comment 14:

Line 121 Please clarify as to what composite samples are. Are they combined samples or just samples of each food individually?

Authors' response 14:

Line 149-150: The following phrase was added to explain what composite samples mean.

“(a mixture of individual samples)”

Reviewer 2 comment 15:

Line 128 "gram of portion per day" is not clear. Should is just be "grams per day"?

Authors' response 15:

Line 156: corrected as follows.

“recommended portion per day”

Line 158: corrected as follows.

“recommended portion per day”

Reviewer 2 comment 16:

Were t-test considered to compare means between groups?

Authors' response 16:

We did not understand this comment. We did not conduct a t-test to compare means between groups.

Reviewer 2 comment 17:

Results:

Line 185 What is "organ" meat?

Authors' response 17:

Line 138; Line 221: organ meat is deleted from the manuscript as it is not included in the FAO & FHI 360 (2016)´s ten food groups.

Reviewer 2 comment 18:

Line 190-191 sentence beginning "This study.." is not presenting results and should be removed.

Authors' response 18:

Line 231-232: The following sentence is deleted:

“This study showed the distribution of dietary diversity over different influencing factors.”

Reviewer 2 comment 19:

Discussion: Line 215-216 is not clear and needs to be reworked.

Authors' response 19:

Lines 306-309: The sentence was revised in order to improve clarity.

“…food intake habit (quantity, quality and diversity) during lactation, which is in contrast to the recommendations in the essential nutrition actions…”

Reviewer 2 comment 20:

Line 222 is not clear.

Authors' response 20:

Line 314: the following phrase is added to explain what media refers to.

“(FM radios and television)”

Reviewer 2 comment 21:

Line 232 could the authors please define what they mean by cash crop

Authors' response 21:

Line 324: The following word is added to explain what cash crop meant in this study’s context: 

“(coffee)”

Reviewer 2 comment 22:

Line 242, incorrect reference format for Ruel

Authors' response 22:

Line 334: “(Ruel, 2003)” changed to “Ruel (2003)”

Reviewer 2 comment 23:

Paragraph 240-244 is not clear e.g. what is "placed in the separate residence"?

Authors' response 23:

Line 332-333: "placed in the separate residence" is replaced with “living in urban and rural residences”

Reviewer 2 comment 24:

Line 245 incorrect reference format for Ajani. Again, this paragraph is not clear.

Authors' response 24:

Line 337: “(Ajani, 2010)” is changed to “Ajani (2010)”

Line 337-340: The paragraph is revised as follows:

“According to Ajani (2010) educational status and household wealth were interrelated and had a positive association with dietary diversity. Higher education attainment is likely to be associated with a higher income and thereby money to buy different food commodity for household food preparation.”

Reviewer 2 comment 25:

Line 253, could the authors please describe what injera is. For example, is it like bread?

Authors' response 25:

Line 349-350: The following phrase is added to describe what Injera is.

“(Ethiopian traditional cereal-based pancake, prepared from fermented cereal sour dough)”

Reviewer 2 comment 26:

Line 257 please indicate what is meant by ash.

Authors' response 26:

Lines 354-355: The following phrase is added to explain what ash content means:

“a measure of the total amount of minerals present within a food”

Reviewer 2 comment 27:

Line 266 has Ca been previously defined (it is assumed this is calcium?)

Authors' response 27:

Line 363: corrected as follows:

“calcium (Ca)”

Reviewer 2 comment 28:

Line 251 - kale sauce has been highlighted specifically in the discussion but has not been described previously. It would be of benefit to describe how kale sauce is used, or what part it plays in the diet prior to this.

Authors' response 28:

Line 345-348: the following sentence is added to describe how kale sauce is used:

“Kale sauce is prepared by cooking onion and oil 4 to 5 minutes and adding chopped and cooked-until-soft kale, and stirring once or twice, for 10 minutes. During our survey we have observed that most households add protein rich ingredients like beans when making kale sauce.”

Reviewer 2 comment 29:

Limitations - appropriate although how representative is this study of the rest of Ethiopia? 

Authors' response 29:

Lines 399-400: The following limitation statement is added as suggested by the reviewer.

“Ethiopia is a very diverse country in terms of food production and consumption culture, and this study may not be representative of the rest of the country.”

Reviewer 2 comment 30:

Strengths - I am unclear how this is a strength. This is an outcome from the study but the methodology undertaken to achieve this is the strength.

Authors' response 30:

Lines 404-405: The sentence was rewritten as follows:

“An official food analysis methodology was employed to analyze the composition of undocumented maternal diets.”

Reviewer 2 comment 31:

Conclusions:

Line 307-308 this sentence is not clear and should be reworked.

Authors' response 31:

Lines 410-412: The sentence is revised as follows:

“Kale sauce, which is traditionally popular in South and Southwest part of rural Ethiopia, tend to provide relatively healthy macro and micronutrients to the lactating mothers.”

Reviewer 2 comment 32:

Table 4 - why was 0.01 used as the p value when .05 is the standard and this is stated in line 147.

Authors' response 32:

Line 246: corrected as suggested by the reviewer.

Reviewer 2 comment 33:

Other tables appropriate although only descriptive in nature

Authors' response 33:

Lines 257-263: A new table is added:

“Table 5. Multivariable logistic regression model predicting the likelihood of having Minimum Dietary diversity (MDD-W) among women in Jimma Zone, Southwest Ethiopia”

Reviewer #3: 

Reviewer 3 comment 1:

While the topic of the study has public health importance, measurement approaches, the tools and the data analysis is not scientifically sounding. Despite the result section and conclusion attempted to note the variables associated with low dietary diversity there is no a strong method of data analysis to test the hypothesis. The study used simple chi-square which simply allows us to see the presence or absence of any association between two variables at a time. This does not help us to know which variable is cause and which is an effect? In what magnitude? In what direction (favoring or not).

Authors' response 1:

Line 174-175:

“Chi-square test was used to see the association between dietary diversity and socio-economic and demographic variables.” 

was changed to 

“Bivariate and multivariate logistic regression statistical analysis was carried out to determine the factors associated with diet diversity in lactating mothers.”

Lines 249-271: The following results of multivariate logistic regression were added:

“On multivariable logistic regression analyses presented in Table 5, after adjusting for different variables, meeting Minimum dietary diversity (MDDS) of women was positively associated with agricultural production diversity(P=0.001) and grade of the women(P=0.04). For a unit increase in agricultural production diversity of the household, the likelihood of achieving minimum dietary diversity of women was 1.4 times higher (AOR: 1.394, 95%CI: 1.142, 1.701). Similar, for one grade increase in the educational status of women the odds of achieving minimum, dietary diversity was 1.5 times higher( AOR: 1.485, 95%CI: 1.018, 2.168).

Table 5. Multivariable logistic regression model predicting the likelihood of having Minimum Dietary diversity (MDD-W) among women in Jimma Zone, Southwest Ethiopia

Model B P AOR 95% C.I.

Lower Upper

District 

Manna -0.211 0.599 0.810 0.369 1.779

Omo-nada -1.444 0.003 0.236 0.090 0.616

Dedo 1.000 

Food Production diversity 0.332 0.001 1.394 1.142 1.701

Grade of the women 0.396 0.040 1.485 1.018 2.168

Wealth Tertitle 

Poor -0.732 0.154 0.481 0.176 1.317

Medium -1.013 0.062 0.363 0.125 1.052

Rich 1.000 

Age of the woman(yrs) -0.065 0.135 0.937 0.861 1.020

Household has alternative income source -0.175 0.639 0.839 0.404 1.744

Place of residence, 

Rural -1.263 0.019 0.283 0.099 0.810

Urban 1.00 

Educational status of the husband 0.894 0.085 2.446 0.883 6.776

Household size 0.150 0.171 1.162 0.937 1.441

Production diversity: a score generated by summing the number of animal and plant source food that the household reported to produce. 

Maximum Standard error: 0.543, Hosmer Lemeshow Test for Model Fitness (P=0.052).

AOR: Adjusted Odds Ratio.

CI: Confidence Interval.

Conversely, district of the study (P=0.003) and place of residence (P= 0.019 ) were negatively associated with meeting minimum dietary diversity. Women living in Omo-Nada District were 86% less likely to meet the minimum dietary diversity as compared to this who live in Dedo District( AOR: 0.236, 95%CI” 0.090, 0.616). Likewise, women who live in the rural areas were 72% less likely to meet the minimum dietary diversity as compared to those who live in the urban areas (AOR: 0.283, 95%CI: 0.099, 0.810).”

Reviewer 3 comment 2:

Another concern is, construction of survey questionnaire has not been clearly stated and its reliability and validity are not presented. For example, measuring socioeconomic class, educational status etc was not clear.

Authors' response 2:

Line 29-30

“pre-tested and”

Line 121-124: the following paragraph was added to describe how the wealth index was calculated:

“Socio-economic information

A wealth index was determined using principal component analyses based on data on assets encompassing land for agriculture, production of crops, ownership of animals and properties (Garenne & Hohmann-Garenne, 2003). The index was rank divided into tertiles and used for further analyses.”

Line 492-493: The following reference is added to the reference list:

Garenne M, Hohmann-Garenne S (2003) A wealth index to screen high-risk families: Application to Morocco. J Heal Popul Nutr 21:235–242

Reviewer 3 comment 3:

Including different agroecological areas in this study could be taken as one of the strengths. However, it is true that dietary diversity could vary seasonally, and the study has not mentioned the season(s) when the study was undertaken.

Authors' response 3:

Line 107: The following phrase is added to indicate the season of the study:

“post-harvest season which is dry”

Reviewer #4: 

Reviewer 4 comment 1:

Consideration of the nutrient needs of lactating women is an important topic, particularly as longer periods of lactation are being recommended.

Authors' response 1:

Ok

Reviewer 4 comment 2:

An important reference that the authors will want to consider is: FAO and FHI 360. 2016. Minimum Dietary Diversity for Women: a Guide for Measurement. Rome: FAO. The MDD-W has been utilized and validated by several groups. The MDD-W presented in this publication outlines 10 food groups (which are not the same as those specified in this manuscript. And it specifies a cut-off of ≥5 food groups.

Authors' response 2:

We accepted the reviewer's comments and corrected the dietary diversity related information in the manuscript accordingly.

Lines 133-146:

“A single 24-h dietary recall was used to obtain data on dietary diversity. Dietary diversity was assessed with the ten MDD-W food groups namely (1) Grains, white roots and tubers, and plantains, (2) Pulses (beans, peas and lentils), (3) Nuts and seeds (4) Dairy, (5) Meat, poultry and fish, (6) Eggs, (7) Dark green leafy vegetables, (8) Other vitamin A-rich fruits and vegetables, (9) Other vegetables and (10) Other fruits. One point was awarded for the consumption of each food group with a total possible score being 10. Minimum dietary diversity was found to be achieved when a mother consumed mean number of food groups greater than 5 per day and respondents with mean of less than 5 food groups were considered as not achieving the minimum dietary diversity (FAO & FHI 360, 2016).”

Lines 209-263: The dietary diversity results were revised as follows:

Lines 226-227: Table 3 was revised following the FAO & FHI 360, 2016 methodology:

Table 3 Dietary diversity score and percentage of women consuming each of the 9 food groups

 DDS

Mean 3.73

Median 4.0

SD 1.03

Minimum 0

Maximum 8

Minimum Dietary Diversity for Women of Reproductive Age N (%) 

<5 food groups 447 (80.1)

≥5 food groups 111 (19.9)

Consumption N (%) 

Grains, white roots and tubers, and plantains 553 (99.1)

Pulses (beans, peas and lentils) 431 (77.2)

Nuts and seeds 13 (2.3)

Dairy 75 (13.4)

Meat, poultry and Fish 18 (3.2)

Eggs 7 (1.3)

Dark green leafy vegetables 299 (53.6)

Other vitamin A-rich vegetables and fruits 58 (10.4)

Other vegetables 499 (89.4)

Other fruits 129 (23.1)

Lines 230-238: The presence or absence of any association between MDD and socio-economic and demographic variables at a time was revised according to the FAO & FHI 360, 2016 methodology.

“Table 4 shows the association between socio-demographic variables and maternal dietary diversity. Accordingly, dietary diversity score was significantly (P<0.01) affected by districts, residence area, ethnicity, whether the woman is currently attending education or not, husband’s grade and the socio-economic situation of the household. Oromo mothers who live in cereal producing districts, who are not currently attending education, living in poor households and whose husband attended lower grade education were more likely not to achieve MDD as compared with their respective counterparts.”

Lines 244-246: Table 4 was revised accordingly:

Table 4 Distribution of maternal dietary diversity by different variables in Jimma Zone, Southwest Ethiopia

Variables

 Did not achieve MDD Achieved MDD 

 n (%) n (%) P

Districts 

Cash crop 146 (78.5) 40 (21.5) 0.019*

Cereal 161 (86.6) 25 (13.4) 

Vegetable 140 (75.3) 46 (24.7) 

Ethnicity 

Oromo 397 (81.7) 89 (18.3) 0.005*

Amhara 16 (80.0) 4 (20.0) 

Guraghe 6 (75.0) 2 (25.0) 

Tigre 2 (66.7) 1 (33.3) 

Yem 6 (40.0) 9 (60.0) 

Other 20 (76.9) 6 (23.1) 

Educational status of the woman 

1 to 4 76 (78.4) 21 (21.6) 0.134

5 to 8 65 (76.5) 20 (23.5) 

9 to 10 19 (65.5) 10 (34.5) 

Preparatory/TVET 4 (44.4) 5 (55.6) 

Diploma 1 (50.0) 1 (50.0) 

Currently attending education 

No 446 (80.5) 108 (19.5) 0.006*

Yes 1 (25.0) 3 (75.0) 

Husband's grade 

1 to 4 73 (83.9) 14 (16.1) 0.004*

5 to 8 132 (78.6) 36 (21.4) 

9 to 10 26 (57.8) 19 (42.2) 

Preparatory/TVET 17 (89.5) 2 (10.5) 

diploma 2 (50) 2 (50) 

Place of residence 

Rural 297 (79.8) 75 (20.2) 0.822*

Urban 150 (80.6) 36 (19.4) 

Socio-economic status 

Poor 155 (83.3) 31 (16.7) 0.046*

Medium 154 (82.8) 32 (17.2) 

Rich 138 (74.2) 111 (19.9) 

*=Significant at the 0.05 level

Line 496-498: The following reference was added to the list:

“FAO, FHI 360. Minimum Dietary Diversity for Women: A Guide for Measurement [Internet]. Rome: FAO; 2016 Feb [cited 2021 Mar 15]. Available from: http://www.fao.org/documents/card/en/c/cb3434en”

Reviewer 4 comment 3:

There are several inconsistencies in the manuscript that the authors must address. Some, but not all, of these inconsistencies are included in the following points:

L. 22-23 says the average MDD-W was computed from two 24-h recalls, while L. 112 says a single 24-h dietary recall….

Authors' response 3:

Line 30-31: The reviewer is correct. The sentence was corrected accordingly. 

“Minimum diet diversity (MDD-W) was computed from a single 24-h recall.”

Reviewer 4 comment 4:

The nine food groups listed on L. 113 -115 are not the same as the FAO publication nor are they identical to the food groups listed in Table 3 of this manuscript.

Authors' response 4:

Line 134-137: The food groups are revised according to FAO & FHI 360 (201

). 

“(1) Grains, white roots and tubers, and plantains, (2) Pulses (beans, peas and lentils), (3) Nuts and seeds (4) Dairy, (5) Meat, poultry and fish, (6) Eggs, (7) Dark green leafy vegetables, (8) Other vitamin A-rich fruits and vegetables, (9) Other vegetables and (10) Other fruits”

Line 226-227: Table 3 is revised according to FAO & FHI 360 (2016).

Reviewer 4 comment 5:

L. 88-89 says kebeles were sampled from the three selected districts by lottery method. If only by lottery, there could have been 0 to 3 urban kebeles selected from one district. Please clarify the sampling procedure.

Authors' response 5:

Lines 100-104: The sampling method of the kebeles is revised as follows:

“The total kebeles in each woreda were initially stratified into rural and urban areas. Then, three (two rural and one urban) were selected from the existing kebeles based on Probability Proportional to Size (PPS) sampling technique. A total of six rural and three urban kebeles were sampled from the three districts.”

Reviewer 4 comment 6:

L. 127. The document from the Public Health Department of Georgia and World Health Organization (2015) seems to be a report for the general public about healthy eating. Are the recommended grams of portion per day given as dry weight or wet weight. How were these portion sizes adjusted for the calculations in this paper?

Authors' response 6:

We acknowledged, as a limitation of our study that the Public Health Department of Georgia and World Health Organization (2015) report is for the general public about healthy eating and may not directly fit to lactating mothers. The grams of portion per day presented in the Georgia report are given on wet basis. While our food composition results are given on dry weight basis. Because nutrients are found in the dry matter portion of foods, the physical quantity of nutrients will NOT change when water is added or removed.

Reviewer 4 comment 7:

L. 135 mentions the FAO/WHO RNI intakes for mothers. L. 139 specifies division by the RDA. The RNI and the RDA typically are not the same. RDA appears also on Table 5&6.

Authors' response 7:

The definition of RNI used in FAO/WHO report is equivalent to that of recommended dietary allowance (RDA) as used by the Food and Nutrition Board of the US National Academy of Sciences#.

#Food and Nutrition Board, Institute of Medicine. 1997. Dietary Reference Intakes: Washington, DC, National Academy Press.

Reviewer 4 comment 8:

L. 180. Does the Dietary score of zero represent a woman who was fasting??

Authors' response 8:

We suspect a recall bias. There was neither fasting nor feasting when we undergone the survey.

Reviewer 4 comment 9:

L. 210 Fermentation is very likely to increase the bioavailability of Zn, Fe, and perhaps Ca, but usual laboratory analyses and food composition tables show total mineral concentration not bioavailable mineral concentrations.

Authors' response 9:

Line 297: deleted the following phrase as does not fit to the context.

“and fermentation is believed to have played a role in this”

Reviewer 4 comment 10:

L. 222. What is meant by different media?

Authors' response 10:

Line 314: the following phrase is added to explain what media refers to.

“(FM radios, television)”

Reviewer 4 comment 11:

While it is clear that diet diversity was poor and food intakes were inadequate, it is not always clear which differences among the three areas were significantly different from each other. On Table 4, overall differences are shown between “low” and “moderate or high” by chi-square. In L. 224-235 which of the differences described are significantly different when there are more than two categories for a variable?

Authors' response 11:

Line 249-271: The following multivariate analysis result is added to the revised manuscript.

“On multivariable logistic regression analyses presented in Table 5, after adjusting for different variables, meeting Minimum dietary diversity (MDDS) of women was positively associated with agricultural production diversity(P=0.001) and grade of the women(P=0.04). For a unit increase in agricultural production diversity of the household, the likelihood of achieving minimum dietary diversity of women was 1.4 times higher (AOR: 1.394, 95%CI: 1.142, 1.701). Similar, for one grade increase in the educational status of women the odds of achieving minimum, dietary diversity was 1.5 times higher( AOR: 1.485, 95%CI: 1.018, 2.168).

Table 5. Multivariable logistic regression model predicting the likelihood of having Minimum Dietary diversity (MDD-W) among women in Jimma Zone, Southwest Ethiopia

Model B P AOR 95% C.I.

 Lower Upper

District 

Manna -0.211 0.599 0.810 0.369 1.779

Omo-nada -1.444 0.003 0.236 0.090 0.616

Dedo 1.000 

Food Production diversity 0.332 0.001 1.394 1.142 1.701

Grade of the women 0.396 0.040 1.485 1.018 2.168

Wealth Tertitle 

Poor -0.732 0.154 0.481 0.176 1.317

Medium -1.013 0.062 0.363 0.125 1.052

Rich 1.000 

Age of the woman(yrs) -0.065 0.135 0.937 0.861 1.020

Household has alternative income source -0.175 0.639 0.839 0.404 1.744

Place of residence, 

Rural -1.263 0.019 0.283 0.099 0.810

Urban 1.00 

Educational status of the husband 0.894 0.085 2.446 0.883 6.776

Household size 0.150 0.171 1.162 0.937 1.441

Production diversity: a score generated by summing the number of animal and plant source food that the household reported to produce. 

Maximum Standard error: 0.543, Hosmer Lemeshow Test for Model Fitness (P=0.052).

AOR: Adjusted Odds Ratio.

CI: Confidence Interval.

Conversely, district of the study (P=0.003) and place of residence (P= 0.019) were negatively associated with meeting minimum dietary diversity. Women living in Omo-Nada District were 86% less likely to meet the minimum dietary diversity as compared to this who live in Dedo District( AOR: 0.236, 95%CI” 0.090, 0.616). Likewise, women who live in the rural areas were 72% less likely to meet the minimum dietary diversity as compared to those who live in the urban areas (AOR: 0.283, 95%CI: 0.099, 0.810).”

Reviewer 4 comment 12:

A supplementary table is needed to describe the ingredients (recipe) in lentil sauce, bean sauce, pea powder sauce and kale sauce. Particularly for a sauce based on kale, it is hard to imagine a protein concentration of 19.6 g/100 g.

Authors' response 12:

A supplementary file which indicates the type of stews consumed and ingredients of the stews is included. See Supporting information 2.spv.

Line 586: The following caption is added to describe the supporting information files:

“Supporting information 2: Type of stews consumed and ingredients of the stews”

Line 347-348: The following sentence is added as a possible explanation for the high protein content recorded for kale sauce.

“During our survey we have observed that most households add protein rich ingredients like beans when making kale sauce.”

Reviewer 4 comment 13:

There has been work published on phytate and bioavailability (Food & Nutrition Bulletin, 2010) that the authors might find useful.

Authors' response 13:

Line 370: The following word is added:

“(e.g. dephytinization)”

Lines 371-373: the following text was added from the recommended paper.

“combined with enrichment with animal-source foods and/or fortification with appropriate levels and forms of mineral fortificants (Gibson et al., 2010)”

Lines 541-545: The following reference is added to the reference list.

“Gibson, R. S., Bailey, K. B., Gibbs, M., & Ferguson, E. L. (2010). A review of phytate, iron, zinc, and calcium concentrations in plant-based complementary foods used in low-income countries and implications for bioavailability. In Food and Nutrition Bulletin (Vol. 31, Issue 2 SUPPL.). United Nations University Press. https://doi.org/10.1177/15648265100312s206”

---

## [Decision Letter · Decision Letter 1]

4 May 2021

PONE-D-20-19141R1

Maternal dietary practices, dietary diversity, and nutrient composition of Diets of Lactating Mothers in Jimma Zone, Southwest Ethiopia

PLOS ONE

Dear Dr. Forsido,

Thank you for submitting your manuscript to PLOS ONE. After careful consideration, we feel that it has merit but does not fully meet PLOS ONE’s publication criteria as it currently stands. Therefore, we invite you to submit a revised version of the manuscript that addresses the points raised during the review process.

The manuscript has been substantially updated and improved.  However, use of "grammerly" has not resolved all problems.  A careful proofreading is still required to meet the usual PLoS ONE quality. Some minor issues remain that are flagged by the reviewers. These issues should be addressed.

We look forward to receiving your revised manuscript.

Kind regards,

Gideon Kruseman, Ph.D.

Academic Editor

PLOS ONE

Journal Requirements:

Reviewers' comments:

Reviewer's Responses to Questions

**Comments to the Author**

1. If the authors have adequately addressed your comments raised in a previous round of review and you feel that this manuscript is now acceptable for publication, you may indicate that here to bypass the “Comments to the Author” section, enter your conflict of interest statement in the “Confidential to Editor” section, and submit your "Accept" recommendation.

Reviewer #1: All comments have been addressed

Reviewer #2: (No Response)

Reviewer #3: All comments have been addressed

Reviewer #4: (No Response)

2. Is the manuscript technically sound, and do the data support the conclusions?

Reviewer #1: Yes

Reviewer #2: Yes

Reviewer #3: Yes

Reviewer #4: Yes

3. Has the statistical analysis been performed appropriately and rigorously? 

Reviewer #1: Yes

Reviewer #2: Yes

Reviewer #3: I Don't Know

Reviewer #4: I Don't Know

4. Have the authors made all data underlying the findings in their manuscript fully available?

Reviewer #1: Yes

Reviewer #2: Yes

Reviewer #3: Yes

Reviewer #4: Yes

5. Is the manuscript presented in an intelligible fashion and written in standard English?

Reviewer #1: Yes

Reviewer #2: Yes

Reviewer #3: No

Reviewer #4: No

6. Review Comments to the Author

Reviewer #1: - Need to attend to all the comments from the reviewers eg. the reference to cash crop should rather be substituted by the word coffee as it is confusing

Please review the maternal dietary practices paragraph and reword line 119 which is rather deceptive and subject to multiple interpretations.

Reviewer #2: I thank the authors for addressing the previous comments.

Abstract: Some minor issues with English remain (e.g. line 41 "anti-nutritional factors contents" the word factors is probably not required; line 43 should be "the majority").

Introduction: Appropriate.

Methods: Line 92, the reference should perhaps be at the end of the sentence.

Line 95, a brief description of kebele and woreda would be useful to the reader.

Statistical analysis section: univariate and multivariate should be univariable and multivariable

Ethics section: Line 169-170: This should be "The Ethical Review Board agreed to verbal consent as the study was not an intervention"

Line 170, the word "Besides" is not required

Results: Line 197, either dietary diversity score needs (DDS) after it in order to define the abbreviation in the next line or if already defined DDS should be used rather than "dietary diversity score"

Line 220, For clarity "On multivariable logistic regression analyses" should be "Multivariable logistic regression analyses are.." a full stop is required after Table 5 and then the next sentence becomes "After adjusting..

Line 221, if Minimum dietary diversity has been defined, please use the the abbreviation MDD

Discussion: Some minor issues with English remain. e.g. Line 310, "Whereas" is a conjunction and does not start a sentence.

Line 357 "Recall bias. Over and under-reporting of maternal dietary intakes." These are not complete sentences.

Strength of study, while an official food analysis methodology is somewhat of a strength, I think this section does need more consideration. The sampling methodology could also be considered as strength as it was quite rigorous.

Reviewer #3: what does "women in Omo-nada were less likely consuming diversified diet than women in Dedo district" imply? Any policy implication for this finding? Also your analysis showed a year increase in women's age was associated with higher dietary diversity. However, in the discussion you mentioned other studies have found being young age was positively assciated with dietary diversity and this in opposite of your finding.

Reviewer #4: Page Line Comment

3 43-44 This sentence would be much more readable if it said, “ranged between 24.8-65.6 for moisture, 7.6-19.8 for protein,” etc… instead of a long string of numbers followed by a long string of components for the reader to match in order to obtain meaning.

4 46 See above.

6 102 Description of the sampling is improved by adding the urban/rural stratification. Was the “Probability Proportional to Size” technique actually used to select the kebeles?

How does the woreda relate to the district?

8 143-144 Minimum dietary diversity was achieved if a mother consumed five or more food groups per day??

12 199 Consider a different title for this section. Unusual to see insecticide-treated nets and family planning under a heading of dietary practices.

13 208 Table 2 includes more than nutrition

14 226 Table 3 now lists 10 food groups, but the title still says 9 food groups

16 236 Is it statistically valid to compare the 554 women who were not currently in school with the four women who were in school. How large a cell size is required for Chi-Square?

17 Table 4 See question above.

17 Table 4 Add names of districts in parentheses here to make Table 5 meaningful.

17 Table 4 Some ethnicities were merged into other…….Consider if three more are too small for valid chi-square.

18 251 & 254 “grade of the women”……This should be educational level or educational status…..not grade, because individual grade completed is not shown in Table 4.

18 Table 5, L. 252 Food production diversity and agricultural production diversity appear but how they are identified has not been presented in previous tables. How is a unit increase defined? Is it an additional type of crop?

19 274-277 See first comment

20-21 Tables 6 & 7 Clarify if units are /wet_wt or /dry wt somewhere on each of these tables

22 295-298 Unless I missed something in the methodology, proximate analysis of individual foods was conducted. ..most of the foods eaten by women did not contain…

And, …The overall adequacy is less than 1 for each of the food types shown in Table 8.

22 Table 8 MC is in the footnote but not the table.

Table 8 Use Zn in the heading.

25 351 Confirm the RDA for fiber

27 410 The practice of adding beans to kale sauce is mentioned in the paper, but the food product is still referred to as “kale sauce”. To someone who has knowledge of what % protein typically, would be provided by “kale” alone, this is puzzling. Isn’t the addition of beans critical to the protein concentration being reported?

7. PLOS authors have the option to publish the peer review history of their article (what does this mean?). If published, this will include your full peer review and any attached files.

Reviewer #1: No

Reviewer #2: No

Reviewer #3: No

Reviewer #4: No

---

## [Author Response · Author response to Decision Letter 1]

17 Jun 2021

Dear editor, 

We have accepted the critical inputs of the reviewers with great appreciation as it improves our manuscript very much. We have addressed their queries point by point as follows: 

Journal Requirements:

Authors' response:

• We found out that none of the references are retracted.

• For one reference (23) title of the citation is updated.

• Three references (4, 23 & 37) had changed their internet site, which is updated in the current version.

• For nine references (3, 11, 15, 16, 17, 26, 27, 28 and 38) the internet sites where the documents are available.

• Two new references (30, 31) are added to the reference list.

Reviewers' comments:

5. Review Comments to the Author

Reviewer #1: 

Reviewer 1 comment 1:

Need to attend to all the comments from the reviewers eg. the reference to cash crop should rather be substituted by the word coffee as it is confusing

Authors' response 1:

The following corrections were made as suggested by the reviewer.

Line 91: “Cash crop” is changed to “coffee”.

Line 190: Inside Table 1, “cash crop” is changed to “coffee”.

Line 225: Inside Table 4, “cash crop” is changed to “coffee”.

Line 304: “cash crop” is deleted. 

Reviewer 1 comment 2:

Please review the maternal dietary practices paragraph and reword line 119 which is rather deceptive and subject to multiple interpretations.

Authors' response 2:

Line 124-126: The sentence was revised as follows:

“Maternal knowledge and practice in use of insecticide-treated net, consumption of two additional meals in a day, consumption of a variety of food groups, daily use of iodized salt and family planning were assessed (17).”

Reviewer #2: 

Reviewer 2 comment 1:

I thank the authors for addressing the previous comments.

Authors' response 1:

Thank you.

Reviewer 2 comment 2:

Abstract: Some minor issues with English remain (e.g. line 41 "anti-nutritional factors contents" the word factors is probably not required; 

Authors' response 2:

The following corrections were made as suggested by the reviewer:

Line 43: Rewritten as “The content of anti-nutritional factors…”

Line 26: Rewritten as “The content of anti-nutritional factors…”

Line 269: Rewritten as “Table 7 The content of minerals and anti-nutritional factors…”

Reviewer 2 comment 3:

line 43 should be "the majority").

Authors' response 3:

Line 46: “majority” is changed to “most”.

Reviewer 2 comment 4:

Introduction: Appropriate.

Authors' response 4:

Ok

Reviewer 2 comment 5:

Methods: Line 92, the reference should perhaps be at the end of the sentence.

Authors' response 5:

Line 95: The reference is moved to the end of the sentence as suggested by the reviewer.

Reviewer 2 comment 6:

Line 95, a brief description of kebele and woreda would be useful to the reader.

Authors' response 6:

Line 98: The following phrase is added to describe kebele:

“(the smallest administrative unit of Ethiopia)”

Line 98-99: The following phrase is added to describe woreda:

“(the third-level of the administrative division of Ethiopia - after zones and the regional states)”

Reviewer 2 comment 7:

Statistical analysis section: univariate and multivariate should be univariable and multivariable

Authors' response 7:

Line 168: “multivariate” is changed to “multivariable” as suggested by the reviewer.

Reviewer 2 comment 8:

Ethics section: Line 169-170: This should be "The Ethical Review Board agreed to verbal consent as the study was not an intervention"

Authors' response 8:

Line 176-177: Corrected as suggested by the reviewer.

Reviewer 2 comment 9:

Line 170, the word "Besides" is not required

Authors' response 9:

Line 177: The word “Besides” is deleted as suggested by the reviewer.

Reviewer 2 comment 10:

Results: Line 197, either dietary diversity score needs (DDS) after it in order to define the abbreviation in the next line or if already defined DDS should be used rather than "dietary diversity score"

Authors' response 10:

Line 204: “dietary diversity score” is changed to “DDS” as it is already defined in the abstract.

Reviewer 2 comment 11:

Line 220, For clarity "On multivariable logistic regression analyses" should be "Multivariable logistic regression analyses are.." a full stop is required after Table 5 and then the next sentence becomes "After adjusting..

Authors' response 11:

Line 228: Corrected as suggested by the reviewer:

“Multivariable logistic regression analyses are presented in Table 5. After adjusting…”

Reviewer 2 comment 12:

Line 221, if Minimum dietary diversity has been defined, please use the abbreviation MDD

Authors' response 12:

As minimum dietary diversity for women (MDD-W) is defined in the abstract (Line 27), the abbreviation MDD-W is used in the following instances:

Line 33, Line 36, Line 137, Line 140, Line 215 (inside table 3), Line 229, Line 232, Line 234, Line 237-238, Line 247, Line 248, Line 251

Reviewer 2 comment 13:

Discussion: Some minor issues with English remain. e.g. Line 310, "Whereas" is a conjunction and does not start a sentence.

Authors' response 13:

Line 341: Deleted “whereas” and the sentence now begins with “Only unleavened…)

Reviewer 2 comment 14:

Line 357 "Recall bias. Over and under-reporting of maternal dietary intakes." These are not complete sentences.

Authors' response 14:

The incomplete sentences were corrected as follows:

Line 388-389: 

“Recall bias might have occurred as information on maternal dietary practices were collected retrospectively.”

Lines 389:

“The 24-hour recall is prone to over and under-reporting of maternal dietary intakes.”

Reviewer 2 comment 15:

Strength of study, while an official food analysis methodology is somewhat of a strength, I think this section does need more consideration. The sampling methodology could also be considered as strength as it was quite rigorous.

Authors' response 15:

Line 398: As suggested by the reviewer, the following sentence is added as a strength of the study.

“Rigorous sampling methodology was used.”

Reviewer #3: 

Reviewer 3 comment 1:

What does "women in Omo-nada were less likely consuming diversified diet than women in Dedo district" imply? Any policy implication for this finding? 

Authors' response 1:

Line 311-316: The following discussion is added to address reviewer’s comments:

“The variation in dietary diversity between vegetable and cereal producing districts indicates that households tend to consume what they have produced; perishables (vegetables) are more likely to be consumed by the producers than durables (cereals) as they spoil fast if not sold. Therefore, increasing the diversity of farm should be encouraged as one key strategy to improve diet diversity and quality (30).”

Reviewer 3 comment 2:

Also your analysis showed a year increase in women's age was associated with higher dietary diversity. However, in the discussion you mentioned other studies have found being young age was positively associated with dietary diversity and this in opposite of your finding.

That discussion was presented wrongly in the previous submission. It has been corrected as follows in the current submission:

Line 302-304: “Regarding the association between maternal age and DDS, a contradicting report has been documented from Nigeria where being young age was positively associated with dietary diversity (24).”

Reviewer #4: 

Reviewer 4 comment 1:

3 43-44 This sentence would be much more readable if it said, “ranged between 24.8-65.6 for moisture, 7.6-19.8 for protein,” etc… instead of a long string of numbers followed by a long string of components for the reader to match in order to obtain meaning.

Authors' response 1:

Lines 37-40: Corrected as suggested by the reviewer.

Reviewer 4 comment 2:

4 46 See above.

Authors' response 2:

Lines 41-42: corrected as suggested by the reviewer.

Reviewer 4 comment 3:

6 102 Description of the sampling is improved by adding the urban/rural stratification. Was the “Probability Proportional to Size” technique actually used to select the kebeles?

Authors' response 3:

Line 101-102: This was a mistake and is corrected in the revised version as follows:

“probability non-proportional to size”

Reviewer 4 comment 4:

How does the woreda relate to the district?

Authors' response 4:

Line 98: The following text is added to show how woreda is related to district.

“district/woreda”

Reviewer 4 comment 5:

8 143-144 Minimum dietary diversity was achieved if a mother consumed five or more food groups per day??

Authors' response 5:

Lines 137-138: Corrected as indicated by the reviewer.

Reviewer 4 comment 6:

12 199 Consider a different title for this section. Unusual to see insecticide-treated nets and family planning under a heading of dietary practices.

Authors' response 6:

Line 192: As advised by the reviewer, the section title is revised to “Maternal practice of essential nutrition actions”.

Line 122: Similar modification is done for the section title.

Line 285: Similar modification is done for the section title.

Reviewer 4 comment 7:

13 208 Table 2 includes more than nutrition

Authors' response 7:

Line 201-202: The caption is revised as follows:

“Maternal practice of essential nutrition actions during lactation in three districts of Jimma Zone, Southwest Ethiopia”

Reviewer 4 comment 8:

14 226 Table 3 now lists 10 food groups, but the title still says 9 food groups

Authors' response 8:

That was an error. It is corrected.

Reviewer 4 comment 9:

16 236 Is it statistically valid to compare the 554 women who were not currently in school with the four women who were in school. How large a cell size is required for Chi-Square?

Authors' response 9:

The reviewer is correct. “The Chi-square test is invalid if we have fewer than 5 observations in a cell”. Therefore, we have deleted, from Table 4, the finding presenting whether the mother is currently attending education or not.

Line 220: deleted “whether the woman is currently attending education or not,”

Line 222: deleted “who are not currently attending education,”

Reviewer 4 comment 10:

17 Table 4 See question above.

Authors' response 10:

Line 225-226: We have deleted, from Table 4, the finding presenting whether the mother is currently attending education or not.

Reviewer 4 comment 11:

17 Table 4 Add names of districts in parentheses here to make Table 5 meaningful.

Authors' response 11:

Names of the districts were put in a more meaningful fashion as follows.

Coffee producing (Mana)

Cereal producing (Omo Nada)

Vegetable producing (Dedo)

Reviewer 4 comment 12:

17 Table 4 Some ethnicities were merged into other…….Consider if three more are too small for valid chi-square.

Authors' response 12:

As suggested by the reviewer, the fourth ethnicity “Tigre” was added to “Others” because it would be too small for a valid chi-square.

Reviewer 4 comment 13:

18 251 & 254 “grade of the women”……This should be educational level or educational status…..not grade, because individual grade completed is not shown in Table 4.

Authors' response 13:

Line 34, Line 221, Line 223, Line 226 (inside Table 4), Line 230, Line 233, Line 239 (inside table 5): We have changed “grade” with “educational level” for the women as well as their husbands. 

Reviewer 4 comment 14:

18 Table 5, L. 252 Food production diversity and agricultural production diversity appear but how they are identified has not been presented in previous tables. How is a unit increase defined? Is it an additional type of crop?

Authors' response 14:

Line 240-241: At the footnote of Table 5, we have indicated how the production diversity score is generated. Production diversity is a score generated by summing the number of animal and plant source food that the household reported to produce. A unit increase in this context can be defined as an additional type of animal or plant source food produced by the household. 

Reviewer 4 comment 15:

19 274-277 See first comment

Authors' response 15:

Line 255-260: Corrected as suggested by the reviewer.

Reviewer 4 comment 16:

20-21 Tables 6 & 7 Clarify if units are /wet_wt or /dry wt somewhere on each of these tables

Authors' response 16:

Line 263: The following information is added to the Table 6 caption:

“(dry weight basis)”

Line 269: The following information is added to the Table 7 caption:

“(dry weight basis)”

Reviewer 4 comment 17:

22 295-298 Unless I missed something in the methodology, proximate analysis of individual foods was conducted. ..most of the foods eaten by women did not contain…

And, …The overall adequacy is less than 1 for each of the food types shown in Table 8.

Authors' response 17:

As indicated by the reviewer, proximate, mineral and anti-nutrient contents analysis of individual foods was conducted. 

Line 275-276: “…did not contain...” is changed to “…do not provide the recommended daily allowances for…”

Line 278-279: The following phrase is added to elaborate what MAR<1 means:

“…which indicates the requirements for all the nutrients were not met.”

Reviewer 4 comment 18:

22 Table 8 MC is in the footnote but not the table.

Authors' response 18:

MC=Moisture content is deleted from the footnote of Table 8 as it happened to be there by mistake.

Reviewer 4 comment 19:

Table 8 Use Zn in the heading.

Authors' response 19:

Corrected.

Reviewer 4 comment 20:

25 351 Confirm the RDA for fiber

Authors' response 20:

According to the American Heart Association, the daily value for fiber for women under 50 is 21 to 25 grams per day. The value in Table 6 is revised accordingly.

Reviewer 4 comment 21:

27 410 The practice of adding beans to kale sauce is mentioned in the paper, but the food product is still referred to as “kale sauce”. To someone who has knowledge of what % protein typically, would be provided by “kale” alone, this is puzzling. Isn’t the addition of beans critical to the protein concentration being reported?

Authors' response 21:

This is a legitimate concern. To avoid the confusion, we renamed the “kale sauce” to “kale-bean sauce” in Tables 6, 7 and 8.

Reviewer # 5

Reviewer 5 comment 1:

Significant revision were made and the revised manuscript is more appropriate and ready for publication. I have some concerns, which I stated as follows.

1. The two versions of abstract (abstract copied and pasted in the manuscript submission system and abstract in the main manuscript) are not the same. 

Authors' response 1:

This is corrected.

Reviewer 5 comment 2:

2. Sample recruitment period in the original manuscript was changed from March to May 2014 to February 2014 (one month back). Why did the authors change on the period of sample recruitment? 

Authors' response 2:

March to May 2014 is when we collected the data. The preliminary tasks were completed ahead of the data collection.

Reviewer 5 comment 3:

3. Tools (DD) in the original was 9 but changed to ten. This has to be clarified.

Authors' response 3:

This was suggested by the reviewer during the first revision, and we agreed with the reviewer’s comment. The FAO, FHI 360 (2016) guide for measurement of Minimum Dietary Diversity for Women (MDD-W) is the most applicable technique for the study population. 

Reviewer 5 comment 4:

Result

4. Table 5: Model adequacy is marginally significant (p=0.052) suggesting the model does not correctly fit. For which variable does maximum standard error of 0.543 mean?

Authors' response 4:

We expected Hosmer-Lemeshow test to be non-significant (P>0.05) for fitness of the model based on literature. The value we got for Hosmer-Lemeshow test in this analysis was non- significant (P=0.052). Our conclusion therefore is the model is fit. 

Large standard error is an indicator of multicollinearity as approximately 95% of the observations should fall within plus/minus 2 standard error of the regression from the regression line. So, we used Standard error of > 2 as an indicator of multicollinearity and checked it for all variables in the multivariable regression output. However, all variables had low standard error with the maximum standard error being. 0.543 (for educational status of the husband) showing the absence of multicollinearity.

Reviewer 5 comment 5:

5. Two important findings need further explanation. The first finding is women who had increased agricultural productivity had better dietary diversity. Another finding is urban women were better in DD than rural women. It may be true to say agriculture activity is more common in rural than urban. Thus, from these statements, one can assume rural women could be better in DD than urban because of increased agricultural productivity. How do authors explain these controversial findings?

Authors' response 5:

Line 319-327: The following explanation is added to explain the urban-rural disparity in DD.

“Considering the fact that rural areas are hubs for agricultural production, one can assume rural women could be better in dietary diversity than urban counterparts. However, production diversification may not always mean dietary diversity. For example, a study in rural Nigeria found out that production diversification has no statistically significant effect on the dietary diversity of households (30). Other researchers also argue that despite theoretical basis for the correlation between production diversity and dietary diversity, “there is a need for a deeper empirical understanding of how, under what circumstances, and through what pathways own-production of nutritious foods improves diets” (31).”

The following new references were added to the reference list:

Line 525-528: 

“Ayenew HY, Biadigilign S, Schickramm L, Abate-Kassa G, Sauer J. Production diversification, dietary diversity and consumption seasonality: Panel data evidence from Nigeria. BMC Public Health [Internet]. 2018 Aug 8 [cited 2021 Jun 15];18(1):1–9. Available from: https://doi.org/10.1186/s12889-018-5887-6”

Line 529-531: 

“Aberman N-L, Roopnaraine T. To sell or consume? Gendered household decision-making on crop production, consumption, and sale in Malawi. Food Secur [Internet]. 2020;12(2):433–47. Available from: https://doi.org/10.1007/s12571-020-01021-2”

Reviewer 5 comment 6:

Discussion: 

6. Line 309 (…we have observed that most households add protein rich ingredients…). Nothing was said in methods section to clarify this study also involved observation. 

Authors' response 6:

We didn’t do observational study, but we have collected food samples from several households. We have revised the sentence to this context.

Line 337: “…our survey…” is changed to “…collection of food samples,…” 

Reviewer 5 comment 7:

7. Line 338, it is uncommon to write like this (“According to (34) …). Instead, it would be more informative to re-write what was said by another study). 

Authors' response 7:

Line 367-368: Corrected as suggested by the reviewer:

“The ideal cut-off for nutrient adequacy should be 1, which would mean that all the nutrients were consumed in a sufficient amount (36).”

Reviewer 5 comment 8:

Strength:

8. It is not clear what does “an official food analysis methodology” mean? This was not described elsewhere. 

Authors' response 8:

Line 147: “standard methods” was changed to “official methods”

Thank you

---

## [Editor Report · Decision Letter 2]

24 Jun 2021

Maternal dietary practices, dietary diversity, and nutrient composition of Diets of Lactating Mothers in Jimma Zone, Southwest Ethiopia

PONE-D-20-19141R2

Dear Dr. Forsido,

We’re pleased to inform you that your manuscript has been judged scientifically suitable for publication and will be formally accepted for publication once it meets all outstanding technical requirements.

Kind regards,

Gideon Kruseman, Ph.D.

Academic Editor

PLOS ONE
---

## [Editor Report · Acceptance letter]

1 Jul 2021

PONE-D-20-19141R2 

Maternal Dietary Practices, Dietary Diversity, and Nutrient Composition of Diets of Lactating Mothers in Jimma Zone, Southwest Ethiopia 

Dear Dr. Forsido:

I'm pleased to inform you that your manuscript has been deemed suitable for publication in PLOS ONE. Congratulations! Your manuscript is now with our production department. 

Kind regards, 

on behalf of

Dr. Gideon Kruseman 

Academic Editor

PLOS ONE